# Extraordinarily transparent compact metallic metamaterials

Samuel J. Palmer[1], Xiaofei Xiao [1], Nicolas Pazos-Perez[2], Luca Guerrini[2], Miguel A. Correa-Duarte [3], Stefan A. Maier[1,4], Richard V. Craster[5], Ramon A. Alvarez-Puebla [2,6] & Vincenzo Giannini [1,7]

The design of achromatic optical components requires materials with high transparency and low dispersion. We show that although metals are highly opaque, densely packed arrays of metallic nanoparticles can be more transparent to infrared radiation than dielectrics such as germanium, even when the arrays are over 75% metal by volume. Such arrays form effective dielectrics that are virtually dispersion-free over ultra-broadband ranges of wavelengths from microns up to millimeters or more. Furthermore, the local refractive indices may be tuned by altering the size, shape, and spacing of the nanoparticles, allowing the design of gradient-index lenses that guide and focus light on the microscale. The electric field is also strongly concentrated in the gaps between the metallic nanoparticles, and the simultaneous focusing and squeezing of the electric field produces strong 'doubly-enhanced' hotspots which could boost measurements made using infrared spectroscopy and other non-linear processes over a broad range of frequencies.

[1] The Blackett Laboratory, Imperial College London, London SW7 2AZ, UK. [2] Department of Physical Chemistry and EMaS, Universitat Rovira i Virgili, 43007 Tarragona, Spain. [3] Department of Physical Chemistry, Singular Center for Biomedical Research (CINBIO), Southern Galicia Institute of Health Research (IISGS), and Biomedical Research Networking Center for Mental Health (CIBERSAM), Universidade de Vigo, 36310 Vigo, Spain. [4] Nanoinstitut München, Faculty of Physics, Ludwig-Maximilians-Universität München, 80539 München, Germany. [5] Department of Mathematics, Imperial College London, London SW7 2AZ, UK. [6] ICREA, Passeig Lluis Companys 23, 08010 Barcelona, Spain. [7] Instituto de Estructura de la Materia (IEM), Consejo Superior de Investigaciones Científicas (CSIC), Serrano 121, 28006 Madrid, Spain. Correspondence and requests for materials should be addressed to S.J.P. (email: samuel.palmer12@imperial.ac.uk) or to X.X. (email: xiaofei.xiao15@imperial.ac.uk) or to V.G. (email: v.giannini@csic.es)

The future of civilisation will be determined by our ability to create new materials: although no material is truly homogeneous, most materials can be characterised by homogeneous macroscopic properties such as refractive index, in which atomistic inhomogeneities smaller than the wavelengths of optical light have been averaged out[1]. Similarly, artificially structured materials known as metamaterials may be described by an effective index, provided that the artificial structuring remains sufficiently subwavelength[2–5].

Early metamaterials included so-called artificial dielectrics, composed of centimeter-scale arrays of metallic particles, that were capable of guiding and focusing radio waves like a dielectric[6]. In contrast to the nanoscale building blocks of modern metamaterials, the metallic particles of early artificial dielectrics were so large compared to the skin depth of the metal that they behaved as perfect conductors and, unsurprisingly, the losses within each particle were negligible such that the arrays were highly transparent to radio waves.

The physics underlying the transparency of densely packed metal metamaterials can be understood by considering how the electrons in metals and dielectrics respond to an electric field (see Fig. 1). In metals, free electrons are driven to the surfaces until the field generated by the build up of surface charges cancels the applied field within the metal (see Fig. 1a). On the other hand, the electrons within dielectrics are bound to their parent molecules or atoms (see Fig. 1b), which polarise in the presence of an electric field[7]. Although the metallic particles comprising the artificial dielectrics possess free electrons, these particles can be regarded as the 'meta-molecules' or '-atoms' as the electrons are only free to move within the confines of the metallic particles (see Fig. 1c), effectively mimicking a dielectric. It is worth noting that this effect is not connected with the creation of plasmonics bands due to the array periodicity[8]. In fact, at optical frequencies the excitation of localised surface plasmon resonances (LSPRs) produces a highly dispersive and lossy metallic response, even for filling fractions as low as 1%[9,10]. Plasmonic coupling produces electric field hotspots between closely packed particles[11], but these suffer from the high losses and narrow bandwidths associated with LSPRs[12]. The effective dielectric presented in Fig. 1a also differs from the case of arrays of dielectric particles with large

permittivities, $\varepsilon \gg 0$, for which localised 'magnetic' plasmon resonances are observed[13].

Recently, there has been an interest in building effective dielectrics for the visible and infrared spectrum using nanoscale metallic particle arrays[14], but the scrutiny of losses has been limited. Chung et al.[15] demonstrated that the effective permittivity and permeability of such arrays can be tuned independently, but assumed long wavelengths without considering losses. Soon after this, effective dielectrics with large, tunable effective indexes were realised experimentally[16,17]. However, the structures in these experiments were not optimised for transparency. Instead, they were only a few layers of nanoparticles thick, such that losses could be largely ignored.

In this work, we show that these artificial dielectrics remain highly transparent to infrared radiation even when the particles are nanoscopic such that the electric field penetrates the particles (which can therefore no longer be considered perfect conductors), and when particles are tightly packed such that there are strong interactions between the particles. In fact, we find that these dense arrays of metallic nanoparticles can actually be more transparent than germanium, renowned for its transparency in the infrared, even for arrays exceeding 75% metal by volume. Despite being composed of highly dispersive metals, the effective dielectrics are actually virtually dispersion-free, enabling the design of optical components that are achromatic throughout the mid-to-far-infrared. Furthermore, the local effective index of these components can be tuned by altering the size, shape, and spacing of the nanoparticles, and is sensitive to the local refractive index of the nanoparticle environment. Within the array, the electric field is enhanced in the gaps between the metallic nanoparticles. By simultaneously exploiting the transparency, tunability, and high metallic filling fraction of the array, we designed a gradient-index lens that both focuses light on the microscale and 'squeezes' the electric field on the nanoscale in order to produce 'doubly-enhanced' electric field hotspots, $|E/E_0|^2 > 10^3$, throughout the infrared. We believe that these hotspots could boost measurements made using infrared spectroscopy and other non-linear processes over a broad range of frequencies and with minimal heat production.

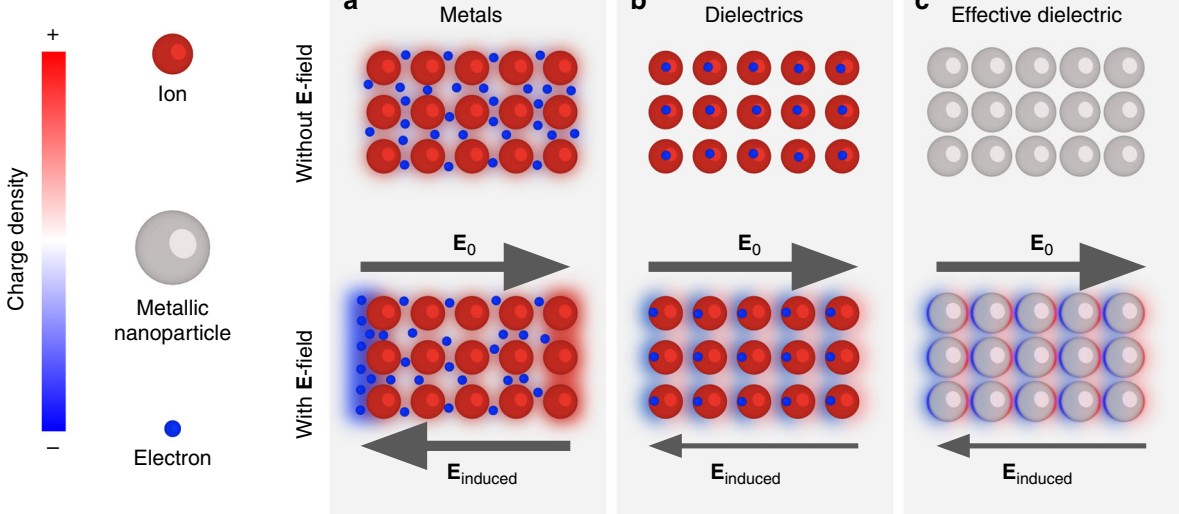

**Fig. 1** An illustration of how metals, dielectrics, and effective dielectrics respond to a slowly varying electric field. Within in each system, the applied field is opposed by an induced electric field generated by the build up of surface charges. **a** In metals, the electrons are free to move until the applied and induced fields cancel in the bulk. In dielectrics (**b**) and effective dielectrics (**c**), the surface charge is generated by the polarisation of the (meta-)atoms or (meta-)molecules, and the induced field is weaker than the applied field

## Results

**Effective behaviour of nanospheres and nanocylinders.** It can be challenging to calculate the effective index of densely packed arrays of metallic nanoparticles. The components are lossy, highly dispersive, and strongly coupled due to the close packing, and conventional methods such as the plane-wave expansion method would suffer from poor convergence. Additionally, there is a large dissimilarity between the length scales of the large wavelength, small particle, and even smaller gaps which require fine meshing. In traditional methods of solving electromagnetic eigenvalue problems, the frequency, $\omega$, is sought as a function of the permittivity and permeability, $\varepsilon$ and $\mu$, at a specific position in the Brillouin zone, $\mathbf{k}$[18],

$$\omega = \omega\Big(\varepsilon(\omega), \mu(\omega), k_x, k_y, k_z\Big). \quad (1)$$

However, this must be solved iteratively as the permittivity and permeability are themselves functions of frequency, which can be particularly problematic for highly dispersive and lossy systems such as ours. Instead of solving for frequency, we solve directly for the Bloch wave vector (see Supplementary Note 1),

$$k_z = k_z\Big(\varepsilon(\omega), \mu(\omega), k_x, k_y, \omega\Big), \quad (2)$$

from which we may extract the effective index, $n_{\text{eff}} = k_z/k_0$ (when $k_x = k_y = 0$). This allows us to solve for the effective index of compact metallic structures with high accuracy, using finite-differences or finite-elements to discretise our system.

Although we solve for the effective index, $n_{\text{eff}} = \sqrt{\varepsilon_{\text{eff}}\mu_{\text{eff}}}$, we can reasonably assume non-magnetic response $\mu_{\text{eff}} = 1$ for the systems and wavelengths considered in this paper (see Supplementary Note 2), such that $\varepsilon_{\text{eff}} \approx n_{\text{eff}}^2$. Note that this assumption is not necessary to comment on the transparency of the system, which is determined by Im[$n_{\text{eff}}$]. It would be necessary to consider the magnetic response for larger particles or smaller wavelengths, such as those described by Chung et al.[15].

In Fig. 2, we contrast the transparency of arrays of nanocylinders and nanospheres to germanium, although in principle the nanoparticles could have other shapes, and demonstrate that these arrays can guide and focus light. Figure 2a

shows that arrays of nanocylinders behave as effective dielectrics ($\varepsilon_{\text{eff}} > 0$) for transverse electric polarised light (TE, red line). This is because a transverse force on the electrons leads to oscillating surface charges that mimic the oscillating dipoles of an atom (or molecule) in a real dielectric. In contrast, the response of the cylinders to transverse magnetic polarised light (TM, blue line) is similar to the response of the bulk metal (grey dashes) as the electrons are free to move under the action of the longitudinal electric field without encountering the surfaces of the cylinders.

Furthermore, it is shown in Fig. 2b that arrays of nanospheres behave as effective dielectrics regardless of the incident polarisation, as the forcing of electrons in any direction results in surface charges that imitate the oscillating dipoles of a dielectric. Remarkably, such arrays can be more transparent than real dielectrics such as germanium, even when the system is over 75% metal as in Fig. 2c.

**Validation of effective index calculations.** To test the accuracy of the proposed theory, a highly ordered colloidal supercrystal[19] was produced with gold nanoparticles of diameter 60 nm (see Supplementary Note 7). The supercrystal was deposited on a germanium substrate (Fig. 3a) and characterised in a UV–vis–NIR spectrophotometer (Fig. 3b). Notably, the material showed an outstanding transparency, demonstrating the feasibility of experimental production of the suggested metamaterials.

Our effective index calculations were also validated using finite-difference time-domain simulations of a plane wave, $\lambda_0 = 2$ μm, impinging on a primitive lens shape from above. As expected, there is no transmission through the solid gold in Fig. 3c because gold has a skin depth of about 13 nm at this wavelength, which is much smaller than the lens thickness of 2 μm. The observed fringes in intensity are in fact caused by diffraction around the lens. However, breaking the solid gold into an array of gold

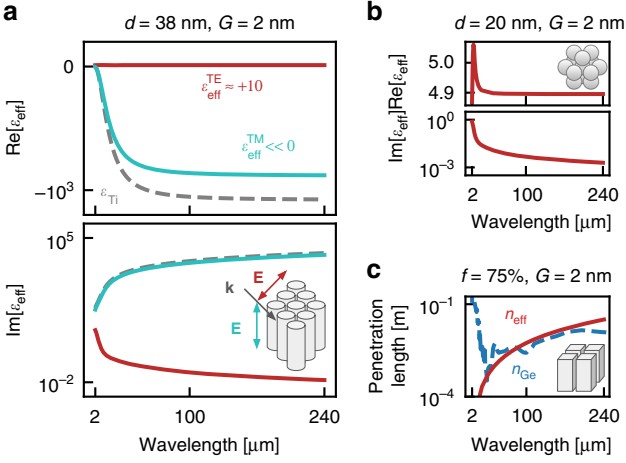

**Fig. 2** Effective permittivity of metallic nanoparticle arrays. **a** The effective permittivity of an array of titanium nanocylinders (with diameter $d = 38$ nm and surface-to-surface separation $G = 2$ nm) for TE (red curve) and TM (blue curve) polarised light compared to the permittivity of solid titanium (dashed curve). **b** The effective permittivity of titanium nanospheres, ($d = 20$ nm, $G = 2$ nm) for unpolarised light. **c** The effective penetration length of the nanoparticle arrays can exceed that of real dielectrics, such as germanium, even for metallic filling fractions as high as 75%

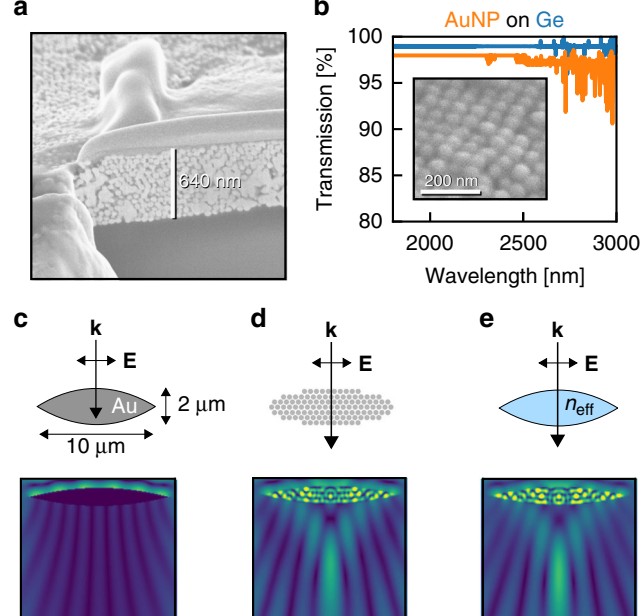

**Fig. 3** Experimental and numerical demonstrations of transparent metallic arrays. **a** Microscopy image of 60 nm diameter gold colloidal supercrystal deposited on a Ge substrate. **b** The metallic particles show high infrared transparency. **c**–**e** The effective dielectrics are transparent enough to act as micrometer-scale lenses to infrared radiation of wavelength $\lambda_0 = 2$ μm, as shown by the magnetic near-fields. There is good agreement between **d** the full geometry of titanium cylinders with diameter 38 nm and surface-to-surface gap 2 nm and **e** the homogenised geometry, $n_{\text{eff}} = 3.2 + 0.5i$

nanocylinders, as in Fig. 3d, produces a transparent lens capable of focusing light which closely resembles the behaviour of the homogeneous dielectric lens of Fig. 3e, despite consisting of 82% metal by volume.

**Comparison of different metals.** As a continuation (see Fig. 4a) we investigate how the effective index and losses are modified by the skin depth, $\delta_s = \lambda_0/4\pi\kappa$, defined as the distance the field penetrates into a metal before the intensity decays by a factor of $1/e$. As shown in the inset of Fig. 4a, different metals have different skin depths, but for Drude metals the skin depth generally increases with wavelength as $\delta_s \sim \sqrt{\lambda}$ throughout the infrared. We observe that metals with longer skin depths produce the most transparent and least dispersive nanoparticle arrays; further discussions regarding the origin of dispersion at near-to-mid infrared wavelengths and losses in the visible and telecom wavelengths can be found in Supplementary Notes 3 and 4, respectively. We note that at long wavelengths the effective index of each array is tending to the same value for all four metals.

Next, we compare our results to the Maxwell Garnett (MG) mixing formula[20–22], in which subwavelength particles are approximated as non-interacting dipoles, and the polarisability of each dipole is taken as the quasi-electrostatic polarisability of the corresponding particle. In principle, this approximation should fail at high filling fractions due to non-dipolar contributions at the small gaps[23–26], but nonetheless remains surprisingly accurate for the filling fractions used here. For cylindrical particles in air or vacuum, the MG mixing formula reads

$$\varepsilon_{\mathrm{MG}}^{\mathrm{cyl}} = \frac{\varepsilon + 1 + f(\varepsilon - 1)}{\varepsilon + 1 - f(\varepsilon - 1)}, \qquad (3)$$

where $\varepsilon$ and $f$ are the permittivity and filling fraction of the cylinders, respectively. When $|\varepsilon| \gg 1$, as with Drude metals at long wavelengths, the MG mixing formula tends to

$$\varepsilon_{\mathrm{MG}}^{\mathrm{cyl}} \approx \frac{1+f}{1-f} - \frac{4f}{(1-f)^2}\frac{1}{\varepsilon} + \mathcal{O}\left(\frac{1}{\varepsilon^2}\right), \qquad (4)$$

which is, to first order, both real and independent of the permittivity.

In Fig. 4b, we investigate the behaviour of metallic nanoparticle arrays when the wavelength ($\lambda_0 = 200\,\mu m$), metal (aluminium), and therefore skin depth (8 nm) are fixed, and the particle sizes and spacings are changed while keeping the filling fraction constant. As expected, the effective index is well approximated by the MG mixing formula when the filling fraction is low and the particles are small compared to the skin depth.

As the sizes of the particles are increased relative to the skin depth, the system diverges from the quasi-electrostatic limit and, at first, the losses rise as electromagnetic field is retarded. However, when the particles become much larger than the skin depth, the losses begin to fall again as the electric field penetrates a decreasing fractional volume of the metal particle. In the limit of $d/\delta_s$ to infinity, this fractional volume tends to zero and the metallic particles behave as if they are made of a perfect conductor (PC, dashes of Fig. 4b). Note that the limiting case of the PC was modelled using the same numerical methods as the regular metals, but applying the boundary condition $\hat{\mathbf{n}} \cdot \nabla H_z = 0$ at the surface of the particles. The PC limit shown in Fig. 4b therefore accounts for the full interactions between particles, whereas the MG limit does not.

At high filling fractions, the MG mixing formula underestimates the effective index of the array. This is because MG neglects the attractive forces between neighbouring surface charges (see Fig. 4b inset) which increase the effective polarisability of the particles.

**Tunability of the effective index.** In addition to being highly transparent, the system is highly tunable by controlling the size, shape, and spacing of the particles. In Fig. 5a, we demonstrate that we can tune an anisotropic response by controlling the aspect ratio of arrays of elliptical cylinders. We compare our numerical method (solid lines) and the corresponding MG mixing formula (dashes),

$$\varepsilon_{\mathrm{MG}}^{\mathrm{ell-cyl}} = \frac{\varepsilon + \Lambda + \Lambda f(\varepsilon - 1)}{\varepsilon + \Lambda - f(\varepsilon - 1)}, \qquad (5)$$

where $\Lambda = L_{\parallel}/L_{\perp}$ is the aspect ratio of the cylinder, and $L_{\parallel}$ and $L_{\perp}$ are the elliptical axes of the cylinder parallel and perpendicular to the electric field, respectively. This expression may be obtained by considering the depolarisation factor of an ellipsoid when one axis is much larger than the other two[27], or by directly solving Laplace's equation in elliptic cylindrical coordinates (see Supplementary Note 6).

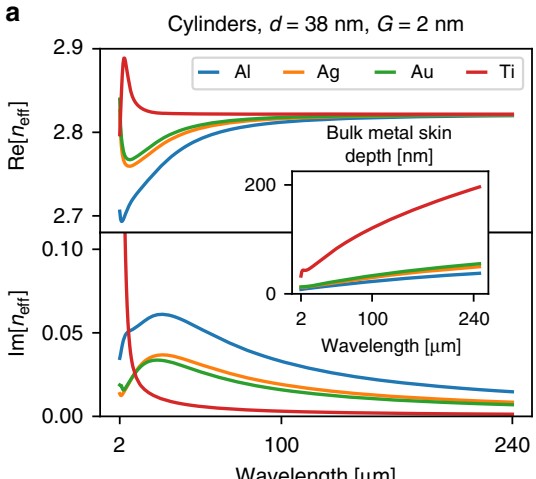

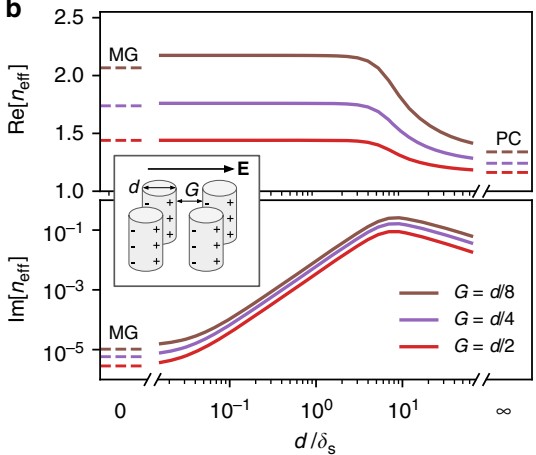

**Fig. 4** Transparency as a function of skin depth. **a** The effective index of a square array of nanocylinders, composed of aluminium, gold, silver, and titanium. Inset: the skin depth of each metal, calculated using the Lorentz–Drude model of permittivity with data from Rakic et al.[29]. **b** At a fixed wavelength, it is the ratio of the particle diameter to the skin depth of the metal that determines whether the particles behave as quasi-static dipoles or perfect conductors. The effective index is remarkably constant for $d \lesssim \delta_s$

The numerical results demonstrate that we can easily tune the effective index to vary by more than 50% as the system is rotated. The geometry, shown in the insets of Fig. 5a, was constructed by distorting square arrays of circular cylinders.

For the real part of $n_{eff}$, there is generally very good agreement between our numerics and the MG mixing formula. However, note that for extreme aspect ratios, $L_\parallel/L_\perp \ll 1$ or $L_\parallel/L_\perp \gg 1$, the real part of $n_{eff}$ is over- and underestimated, respectively, by the MG mixing formula.

From consideration of the geometry shown in Fig. 5a, we believe that interactions between horizontally adjacent particles are enhanced for small aspect ratios, whereas for large aspect ratios it is the interactions between vertically adjacent particles that are enhanced. Since the incident electric field is horizontally polarised, each particle acts as a horizontally aligned dipole, and the effective polarisability of the array is increased by horizontal interparticle interactions and decreased by vertical interparticle interactions.

It is interesting to note that the interparticle interactions are acting to reduce the tunability of the system. This could become significant in other systems, for example, if both the cylinders and the gaps between the cylinders were distorted, then the gaps would become very small at significant aspect ratios and the tunability of the system would be more limited.

The true losses of the distorted array are higher than those predicted by MG, for the same reasons as discussed in Fig. 4b, but are still much lower than for bulk metal. Interestingly, the losses are lower for the distorted arrays as the retardation effects are lessened when the particles are thin.

The effective index can also be tuned by fixing the particle positions and tuning their sizes. In Fig. 5b, there is surprisingly good agreement between the numerics (solid line) and MG formula (dashes). As discussed in Supplementary Note 5, this is because the particle–particle interactions are weakened by the staggered nature of the particles in triangular lattices. While the losses remain low, there is a noticeable increase compared to the MG formula for diameters larger than the skin depth.

**Doubly-enhanced electric field hotspots**. To highlight our ability to tune the local effective index, we constructed a gradient-index (GRIN) lens using the same triangular lattice of gold cylinders that was studied in Fig. 5b, but varying the diameters of the cylinders with position. The schematic of the lens is shown in Fig. 6a. By using the GRIN lens to simultaneously focus light on the microscale and 'squeeze' light on the nanoscale, we were able to produce intense, 'doubly-enhanced' electric field hotspots. Unlike plasmonic enhancements, this effect is not based on lossy resonances and is both broadband and low-loss.

In order to maximise the squeezing of the electric field, the focal point of the GRIN lens must coincide with the region of closest packing. For this reason, we chose the so-called 'concentrator' lens, $n_{eff}^{conc}(r) = R/r$, the focal point of which is located at the origin[28].

The cylinder diameters varied slowly enough that Fig. 5b could still be used to determine the diameter necessary to achieve a given local effective index. In mapping from effective index to cylinder diameter, it was necessary to truncate the effective index of the lens at $n_{eff} \approx 3.5$, as shown in Fig. 6b. This reduced the quality of the focus, but was sufficient for our purposes.

We can see from the magnetic near-fields in Fig. 6c that the performance of the cylinders designed and tuned in this manner agrees well with the targeted effective lens behaviour. Although the lens was tuned to operate at a wavelength of 2 µm, the

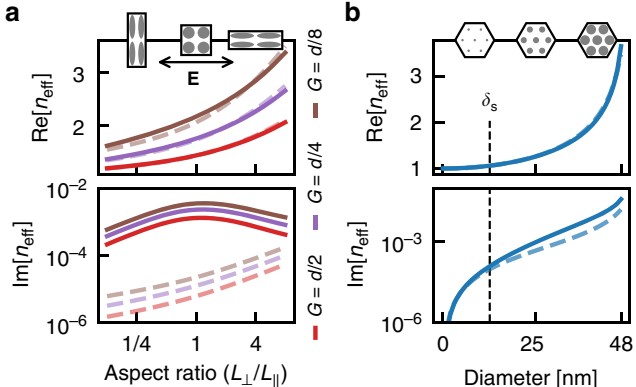

**Fig. 5** The effective index of gold nanocylinders as functions of aspect ratio and particle size. Numerics (solid lines) and Maxwell Garnett mixing formula (dashes). **a** The aspect ratios of square arrays of cylinders were varied, while keeping the volume and surface-to-surface separation of each cylinder constant, as shown in the insets. The undistorted diameter of the cylinders was $d = 30$ nm and the incident wavelength was $\lambda_0 = 200$ µm. **b** The cylinders were placed on a triangular lattice of length 50 nm, and their diameters were varied from 0 nm $\leq d \leq$ 48 nm for an incident wavelength of $\lambda_0 = 2$ µm

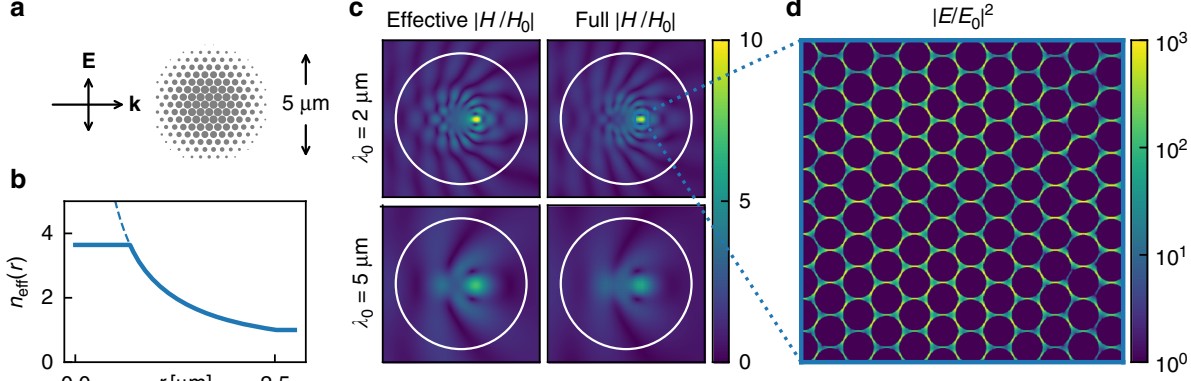

**Fig. 6** Designing a gradient-index lens with 'doubly-enhanced' hotspots. **a** Schematic of a 'concentrator' gradient-index lens composed of gold nanocylinders on a triangular lattice with 50 nm site-to-site separation. **b** Effective index profile of the concentrator lens, ideal (dashed) and achieved (solid). **c** Magnetic near-fields calculated using the effective geometry and the full geometry both confirm that plane waves are focused towards the origin of the lens. **d** Within the focal point of the lens, the combined focusing and squeezing of the electric field produces 'doubly-enhanced' hotspots

dispersion of the metamaterial is low for all wavelengths $\lambda_0 \gtrsim 2$ μm, and we see that the lens performs equally well at larger wavelengths, such as $\lambda_0 = 5$ μm.

Unlike the magnetic field, which is continuous across the air–metal interfaces, the electric field is strongly localised in the gaps as shown in Fig. 6d. This squeezing of a 2 μm wavelength into 2 nm gaps combined with the focusing effects of the lens produced strong hotspots of intensity $|E/E_0|^2 > 10^3$.

## Discussion

Low-loss effective dielectrics were constructed from arrays of metallic nanoparticles. These arrays are highly transparent, at times even exceeding the transparency of real dielectrics renowned for their transparency to low energy radiation, such as germanium, and can be tuned locally by controlling the size, shape, and spacing of the particles.

Furthermore, and in contrast to metamaterials designed upon resonant effects, the effective index is essentially constant for all wavelengths greater than about 2 μm. This allows the design of optical devices to guide or enhance light over an extremely broad range of frequencies, essentially without an upper bound on wavelength. For example, the gradient-index lens presented in Fig. 6 produces strong 'doubly-enhanced' hotspots by simultaneously focusing and squeezing the electric field between the metallic particles. We believe that these hotspots could be used to enhance non-linear processes, such as infrared spectroscopy, over a much broader range of wavelengths and with lower losses than could be achieved with resonance-based enhancements.

## Data availability

Experimental data is available upon reasonable request.

## Code availability

The codes used to generate and plot the data in this article are available at https://github.com/sp94/etcmm, https://doi.org/10.5281/zenodo.2611592.

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

## Acknowledgements

S.J.P. would like to acknowledge his studentship from the Centre for Doctoral Training on Theory and Simulation of Materials at Imperial College London funded by EPSRC Grant No. EP/L015579/1. R.A.A.-P., N.P.-P., L.G., and M.A.C.-D. thank the MINECO-Spain (CTM2014-58481R, CTM2017-84050R, CTQ2017-88648R, RYC-2015-19107 and RYC2016-20331), Xunta de Galicia (Centro Singular de Investigacion de Galicia, Acc. 2016-19 and EM2014/035), Generalitat de Cataluña (2017SGR883), URV (2017PFR-URV_B2-02), URV and Banco Santander (2017EXIT-08) and European Union (ERDF). X.X. acknowledges the Lee Family Scholarship. S.A.M. and R.V.C. acknowledge the EPSRC Mathematical Fundamentals of Metamaterials programme grant (EP/L024926/1) and the US Air Force Office of Scientific Research/EOARD (FA9550-17-1-0300). S.A.M. additionally acknowledges the Lee-Lucas Chair in Physics and DFG Cluster of Excellence Nano-initiative Munich, and the Bavarian "Solar Technologies Go Hybrid" (SolTech) programme. R.V.C. thanks the Leverhulme Trust for their support. V.G. acknowledges the Consejo Superior de Investigaciones Cientficas (INTRAMURALES 201750I039). The authors would also like to thank Rodrigo Berte and Ory Schnitzer for the stimulating discussions.

## Author contributions

S.J.P. produced the calculations of effective index under the supervision of V.G. and R.V.C. X.X. performed the finite-difference simulations of the lenses under the supervision of S.A.M. N.P.-P., L.G., M.A.C.-D. and R.A.A.-P. synthesised and measured the transmission of the experimental samples. S.J.P. wrote the manuscript. V.G. conceived of the project. All the authors contributed to the discussion of the results.

## Additional information

**Competing interests:** The authors declare no competing interests.

