## [Peer Review File · Nature Communications]

Reviewers' Comments:

Reviewer #1:

Remarks to the Author:

The manuscript 'Extraordinarily transparent compact metallic metamaterials' reports about a novel view on metallic metamaterials at long mid-infrared wavelengths, where losses in the metal are sufficiently less than that at visible range, thus a real application of such metamaterials becomes more feasible. In particular, authors demonstrate incredibly high transmission through metamaterial with densely packed metallic nanoparticles (over 75% by volume), which at the same time possess effective refractive index of 3. Last but not least, the electric field of propagating wave is squeezed into tiny gaps, accompanied by its huge enhancement, which can easily find its application within area of non-linear optics (sensing, harmonic generation, etc.) The manuscript is well written, with a high level of English. The idea is clearly presented and discussed. Personally, I believe this can breathe a new life into metamaterials and bring them on the slope of enlightenment at the hype cycle. However, the paper feels to be extremely short and unrevealed – please see my comments below. Each of my comments is minor, but overall I would call it as major revision, so that my comments will not be ignored. I believe the manuscript deserves to be published in Nature Communications after authors respond to my comments:

- 1) Authors solve for the effective index, and then assume a non-magnetic response, $\mu = 1$ (line 109, right column). However, reference [7] of the manuscript claims a change in μ for similar geometry of metamaterial. I wonder if authors could add more discussion about their assumption of non-magnetic response. As for full simulations of the geometry (which is, I assume, simulations of figure 2e,f, and figure 5c), it doesn't convince enough, since agreement in far-field response verifies a proper effective index, and not ϵ and μ separately.
- 2) Figure 2 is overwhelmed with information. It seems that authors wanted to show everything with one figure. For example, there is no information regarding the used metal for figure 2 a-c in the text (it can only be found in either figure legend in b or figure caption). I believe, for the whole figure 2 it is better to use only one type of metal, for example, gold, since the comparison of different metals is in the next figure 3. Secondly, why not use the same geometry for figure 2 a and b?
- 3) Line 153, right column: authors claim detector cut off at 3000 cm^{-1} . However, in figure h the vertical drop of transmission appears at different wavelengths for metamaterial and germanium. Was it shifted intentionally for clarity, or is transparency window of metamaterial is less than detection window? Actually, it would be interesting to see this graph in full size with better resolution and more scale labels in supplementary.
- 4) I wonder why there is a peak in effective index, accompanied by increase in its imaginary part for titanium close to 2 μm wavelength? (it is shown in figure 2c and 3a). Is it because of titanium permittivity? According to the reference [19] used by authors for material properties there is no significant feature at 2 μm wavelength (0.62 eV).
- 5) According to figure 3b, the real part of effective index is reasonably well described with MG mixing formula for $d < 5\delta$. Thus I wonder how well MG formula can agree with curves in figure 3a for the real part. For example, will MG replicate peak for Ti and drops for other metals at low wavelengths? Due to the space limitations, such comparison can be shown in Supplementary.
- 6) Figure 3b: it is completely unclear how the curves were produced. Is it for some particular metal with varied wavelength, or is it for valid for any metal and any wavelength (i.e., skin depth is the only factor for fixed G and d, which influence the effective index)? Finally, how the values for PC were produced?
- 7) Line 227, right column: there should be reference to figure 3b (not 3c, which is absent)
- 8) It would be interesting to compare the performance of round cylinders in square and triangular lattice. According to MG formula, the maximum effective index (i.e., for highest filling ratio, which is for negligibly small gaps) would be larger for triangular lattice. Also I wonder, how well MG formula would describe effective index for triangular lattice, compared to the square lattice.
- 9) Figure 5c: why not add a colorbar scale for magnetic field enhancement, $|H/H_0|$? Then one could split the total $|E/E_0|$ enhancement into effect of focusing and squeezing.
- 10) I wonder, how far into NIR or visible range this effect can work? For example, in figure 3a it

seems that losses for all metals except Ti are still acceptable for low wavelength (though the real part of refractive index is no longer constant). Then why not expand the wavelength range up to telecom (1.5 μm) or even visible range? Especially since this is a simple 2d geometry.

11) I wonder if authors could provide empirical formula for effective mode index for simple geometries, such as round cylinders in square or triangular lattices. Can it be done in explicit form like corrected MG formula?

Reviewer #2:

Remarks to the Author:

This manuscript discusses the manner how a simple lattice of metallic cylinders in dielectric background can mimic a low-loss and low-dispersion dielectric, allowing tunability of optical properties by varying the geometrical structure. The work is mostly theoretical and computational, with a little experimental results. I find the paper rather easy to read, although sometimes the figures are quite fully packed and need some effort from the reader's side in order to be absorbed fully. I still find that the paper contains interesting results. There are, however, some issues (small ones and one more serious) that I am worried about.

1. I understand that the only place where there is experimental discussion is for Figure 2f and 2g. This is not explained understandably enough in the text, which refers to germanium window (what is that?) and it is mentioned that "detector cut off" (what does that mean and how is it connected to the figure?)

2. In Figure 3a different metals are compared against each other. But 3b shows results (probably) for a certain metal. I cannot find explanation which of the four metals is the one which forms the cylinders in Figure 3b.

3. In the end of page 4, the text refers to Figure 3c which does not exist.

4. Figure 3c shows nicely how the Maxwell Garnett agrees fairly well with the simulations in the limiting cases of low and high skin depths. However, for the PC conductor limit, the losses don't match. Why is this? A perfect conductor should be lossless.

5. (The serious one.) Equation (5) generalizes the Maxwell Garnett formula for elliptical cylinders, introducing an extra parameter Λ . Λ is connected to the shape of the ellipse as square root of the axis ratio, and it is derived in the supplementary materials. In my opinion the correct Λ should be simply the axis ratio, not its square root. Correct me if I am wrong: in terms of the depolarization factor of an ellipsoid N (which for a sphere is $1/3$ and a circle $1/2$), this parameter Λ is $(1-N)/N$, and the depolarization factor can be calculated from the axes lengths of the ellipsoid (see Sihvola: Electromagnetic mixing formulas and applications, IET, 1999). For one of the ellipsoid axes being much larger than the other two, N becomes $L_2/(L_1+L_2)$ which leads to Λ being L_1/L_2 , and not its square root.

Manuscript NCOMMS-18-33562A-Z

March 5, 2019

Response to Reviewers

We have answered and addressed in full all of the referee's comments. In this document our responses are in blue, and in the manuscript our changes are in red. A complete list of changes is given at the end.

Reviewer 1

The manuscript 'Extraordinarily transparent compact metallic metamaterials' reports about a novel view on metallic metamaterials at long mid-infrared wavelengths, where losses in the metal are sufficiently less than that at visible range, thus a real application of such metamaterials becomes more feasible. In particular, authors demonstrate incredibly high transmission through metamaterial with densely packed metallic nanoparticles (over 75% by volume), which at the same time possess effective refractive index of 3. Last but not least, the electric field of propagating wave is squeezed into tiny gaps, accompanied by its huge enhancement, which can easily find its application within area of non-linear optics (sensing, harmonic generation, etc.) The manuscript is well written, with a high level of English. The idea is clearly presented and discussed. Personally, I believe this can breathe a new life into metamaterials and bring them on the slope of enlightenment at the hype cycle. However, the paper feels to be extremely short and unrevealed – please see my comments below.

We thank the referee for their very nice comments and suggestions. We agree with all the points and we have modified the manuscript accordingly.

1) Authors solve for the effective index, and then assume a non-magnetic response, $\mu = 1$ (line 109, right column). However, reference [7] of the manuscript claims a change in μ for similar geometry of metamaterial. I wonder if authors could add more discussion about their assumption of non-magnetic response. As for full simulations of the geometry (which is, I assume, simulations of figure 2e,f, and figure 5c), it doesn't convince enough, since agreement in far-field response verifies a proper effective index, and not ϵ and μ separately.

We agree that our assumption of a non-magnetic response could have been better justified. We have added to the supplementary material a study, “Weak magnetic responses”, in which the magnetic near-fields of lenses with the same effective refractive index, n_{eff} , but different effective magnetic responses, μ_{eff} (which determines our choice of $\epsilon_{\text{eff}} = n_{\text{eff}}^2/\mu_{\text{eff}}$), are compared to simulations of the full nanoparticle geometry.

Qualitatively, we can see that the full geometry is most closely matched by the homogenised simulations with a weak magnetic response, $\mu_{\text{eff}} \gtrsim 0.9 < 1$. This study was for a ‘worst case scenario’ of relatively short wavelength ($\lambda_0 = 2 \mu\text{m}$) and relatively large particles ($d = 38 \text{ nm}, G = 2 \text{ nm}$), so we believe that the magnetic responses of all the systems we discuss in the paper will be weak.

Our results appear consistent with reference [7] of the manuscript, the tabulated data in the supplementary material of reference [7] suggests we should expect $\mu_{\text{eff}} \approx 0.996$ for a system of our dimensions.

2) Figure 2 is overwhelmed with information. It seems that authors wanted to show everything with one figure. For example, there is no information regarding the used metal for figure 2 a-c in the text (it can only be found in either figure legend in b or figure caption). I believe, for the whole figure 2 it is better to use only one type of metal, for example, gold, since the comparison of different metals is in the next figure 3. Secondly, why not use the same geometry for figure 2 a and b?

We agree that figure 2 was overloaded with too much information. To remedy this, it has been split into two new figures and made simplifications within each. The main text and figure captions have been updated accordingly.

In the first figure, we have placed the effective index calculations of metallic nanoparticles for different particle shapes and polarisations. As suggested by the reviewer, we now use the same metal (titanium) for each system.

The second figure then validates the results of the previous figure by studying the nanoparticles under illumination, and contains both the simulated lens near-fields and the experimental transmission measurements. For simplicity, both the experiment and simulations in this figure now use gold nanoparticles.

3) Line 153, right column: authors claim detector cut off at 3000 cm^{-1} . However, in figure h the vertical drop of transmission appears at different wavelengths for metamaterial and germanium. Was it shifted intentionally for clarity, or is transparency window of metamaterial is less than detection window? Actually, it would be interesting to see this graph in full size with better resolution and more scale labels in supplementary.

As the referee appreciates, the data was shifted for clarity. However, based on the reviewers comments this is unnecessarily confusing and we have re-plotted the data for wavelengths within the detector range only (now figure 3b). We have also included higher resolution experimental images in the supplementary information.

4 and 5) I wonder why there is a peak in effective index, accompanied by increase in its imaginary part for titanium close to $2\ \mu\text{m}$ wavelength? (it is shown in figure 2c and 3a). Is it because of titanium permittivity? According to the reference [19] used by authors for material properties there is no significant feature at $2\ \mu\text{m}$ wavelength (0.62eV).

According to figure 3b, the real part of effective index is reasonably well described with MG mixing formula for $d < 5\delta$. Thus I wonder how well MG formula can agree with curves in figure 3a for the real part. For example, will MG replicate peak for Ti and drops for other metals at low wavelengths? Due to the space limitations, such comparison can be shown in Supplementary.

Figure 1: (a) Effective index of square arrays of metallic cylinders, calculated using the Maxwell Garnett mixing formula for different wavelengths. (b) When the particles are small compared to the skin depth, the full wave simulations of the effective index are qualitatively similar to those of MG, although the effective index is raised by the particle-particle interactions. (c) As the particles become larger, the effective index drops at smaller wavelengths where the skin depth is large compared to the particle diameter and the quasi-static approximation breaks down.

The peak of the effective index of the titanium in figure 3a (now figure 4a, and also as figure 1 in this document) at $5\ \mu\text{m}$ is indeed a property of the permittivity of titanium, and is replicated by the Maxwell Garnett mixing formula (see figure 1a of this document). Specifically, titanium is a poor metal at these wavelengths (ϵ is not very large or negative compared to the other metals).

In contrast, the drop in effective index that is seen for the other metals is not replicated by the Maxwell Garnett formula. This is because the drop is caused by a breakdown of the quasi-static approximation as the skin depth of the metals shrinks with decreasing wavelength; this drop is consistent with the behaviour seen in figure 3b (now figure 4b). This is demonstrated by figures 1b,c of this document, where the dip in effective index is bigger for larger particles. The dip is not seen for titanium as the skin depth of titanium is significantly larger than the other metals, even at small wavelengths.

This information has been added to a new section in the supplementary material, “Origins of the dispersion at near-to-mid infrared wavelengths.”

6) Figure 3b: it is completely unclear how the curves were produced. Is it for some particular metal with varied wavelength, or is it for valid for any metal

and any wavelength (i.e., skin depth is the only factor for fixed G and d , which influence the effective index)? Finally, how the values for PC were produced?

We agree that it was unclear how figure 3b (now figure 4b) was produced. We now explain in the main text and the figure caption that the wavelength ($\lambda_0 = 200 \mu\text{m}$), metal (aluminium), and therefore skin depth (8 nm) are fixed, and the particle sizes and spacings are changed while keeping the filling fraction constant. We have also added a sentence in the main text to explain how we modelled particles as perfect conductors by applying the boundary condition $\hat{\mathbf{n}} \cdot \nabla H_z = 0$ at the surface of the particles.

7) Line 227, right column: there should be reference to figure 3b (not 3c, which is absent)

Agreed, this reference should have been to figure 3b (now figure 4b).

8 and 11) It would be interesting to compare the performance of round cylinders in square and triangular lattice. According to MG formula, the maximum effective index (i.e., for highest filling ratio, which is for negligibly small gaps) would be larger for triangular lattice. Also I wonder, how well MG formula would describe effective index for triangular lattice, compared to the square lattice.

I wonder if authors could provide empirical formula for effective mode index for simple geometries, such as round cylinders in square or triangular lattices. Can it be done in explicit form like corrected MG formula?

This is an interesting question, and we have now addressed it in the supplementary information. In brief, the dipole-dipole interactions are stronger for cylinders on the square lattice, as the dipoles of the cylinders are more directly aligned. These interactions increase the effective polarisability of the cylinders and, as they are neglected by the MG mixing formula, we find that MG underestimates the effective index of the square lattice. In contrast, Maxwell Garnett accurately predicts the effective index of the triangular lattice to within about 2%, even for high filling fractions $f > 80\%$.

The Rayleigh formula is an extension of the MG mixing formula that takes into account particle-particle interactions, up to a certain order. For cylinders in a square lattice, the Rayleigh formula is [Henrik Wallén, Henrik Kettunen, and Ari Sihvola. “Mixing formulas and plasmonic composites.” *Metamaterials and plasmonics: fundamentals, modelling, applications*. Springer, Dordrecht, 2009. 91-102]

$$\epsilon_{\text{eff}} = 1 + \frac{2f}{\frac{\epsilon_r+1}{\epsilon_r-1} - f - \frac{\epsilon_r-1}{\epsilon_r+1} (0.3058f^4 + 0.0134f^8)}. \quad (1)$$

In principle, a similar formula could be found for cylinders on a triangular lattice, but as we have seen in figure 2 of this document, MG is sufficiently accurate for our purposes, especially for cylinders on a triangular lattice, and we feel that the Rayleigh formula wouldn’t offer further insight into the physics.

This information has been added to a new section in the supplementary material, “Accuracy of the Maxwell-Garnett mixing formula for square and triangular lattices”.

Figure 2: The effective indices of metallic cylinder arrays are compared to the Maxwell Garnett mixing formula (blue curve) and the Rayleigh formula (orange curve) for both square and triangular lattices (bottom right insets), with a fixed lattice parameter $L = 40 \text{ nm}$. The filling fraction is varied, up to a minimum surface-to-surface separation of 2 nm . The cylinders are made of gold and the wavelength is $\lambda_0 = 2 \mu\text{m}$.

9) Figure 5c: why not add a colorbar scale for magnetic field enhancement, $|H/H_0|$? Then one could split the total $|E/E_0|$ enhancement into effect of focusing and squeezing.

We have now included a colorbar scale for figure 5c (now figure 6c). Previously the color plots were clamped at $H/H_0 < 5$, so we have also raised the upper limit to $H/H_0 < 10$.

10) I wonder, how far into NIR or visible range this effect can work? For example, in figure 3a it seems that losses for all metals except Ti are still acceptable for low wavelength (though the real part of refractive index is no longer constant). Then why not expand the wavelength range up to telecom ($1.5 \mu\text{m}$) or even visible range? Especially since this is a simple 2d geometry.

We agree that losses in the telecom wavelengths would be of particular interest to many readers, and we have therefore added a new section to the supplementary material, “Losses in the telecom and visible wavelengths”, which includes a plot of the losses from $0.5 \mu\text{m} \leq \lambda_0 \leq 2 \mu\text{m}$. In summary, we find that at $\lambda_0 = 1.5 \mu\text{m}$, the effective skin depth is on the order of a micron, which may be sufficiently transparent for sub-micron devices. We also remark on the behaviours of the different metals, and use equation 4 of the main text to ex-

Figure 3: The imaginary part of the effective index of metallic nanocylinders on a triangular lattice, in the telecoms and visible range. The particles have diameter $d = 38$ nm and surface-to-surface separation $G = 2$ nm.

plain that titanium has the highest losses despite having the largest skin depth of the four metals because the permittivity of titanium is relatively small at these wavelengths.

Reviewer 2

This manuscript discusses the manner how a simple lattice of metallic cylinders in dielectric background can mimic a low-loss and low-dispersion dielectric, allowing tunability of optical properties by varying the geometrical structure. The work is mostly theoretical and computational, with a little experimental results. I find the paper rather easy to read, although sometimes the figures are quite fully packed and need some effort from the reader's side in order to be absorbed fully. I still find that the paper contains interesting results. There are, however, some issues (small ones and one more serious) that I am worried about.

We thank the reviewer for their response. We have updated our manuscript to address all of their comments.

1. I understand that the only place where there is experimental discussion is for Figure 2f and 2g. This is not explained understandably enough in the text, which refers to germanium window (what is that?) and it is mentioned that “detector cut off” (what does that mean and how is it connected to the figure?)

In this context, germanium window simply means germanium substrate, so we have changed ‘window’ to ‘substrate’ in the text.

The ‘detector cut off’ refers to the end of the operating band of the Peltier-cooled PbS detector that was used in the experiment. However, we have realised it is unnecessarily confusing to show data from outside of the detector range, so we have re-plotted the data for wavelengths $1800 \text{ nm} \lesssim \lambda_0 \lesssim 3000 \text{ nm}$ (now figure

3b), and removed the discussion of the cut off from the main text.

2. In Figure 3a different metals are compared against each other. But 3b shows results (probably) for a certain metal. I cannot find explanation which of the four metals is the one which forms the cylinders in Figure 3b.

The reviewer is correct that we neglected to mention that the cylinders in figure 3b (now figure 4b) were made of aluminium. This has been added to the caption and the main text, and we also note that the conclusions should be independent of the specific metal used.

3. In the end of page 4, the text refers to Figure 3c which does not exist.

Agreed, this reference should have been to figure 3b (now figure 4b).

4. Figure 3c shows nicely how the Maxwell Garnett agrees fairly well with the simulations in the limiting cases of low and high skin depths. However, for the PC conductor limit, the losses don't match. Why is this? A perfect conductor should be lossless.

The reviewer is correct that the losses should be zero in the limit of a perfect conductor. We have amended the main text to clarify that the solid lines of figure 3b (now figure 4b) are approaching, but do not reach, the limit of a perfect conductor ($d/\delta_s \rightarrow \infty$). For further emphasis, we have also introduced a visual separation between the limiting cases (dashed lines) and the solid lines by adding breaks to the x -axis.

Ideally, we would wish for this figure to extend to larger values of d/δ_s , but simulating particles hundreds of times larger than the skin depth becomes computationally expensive because the mesh density at the surface of the particles must be on the order of the skin depth. However, we believe the figure shows convincingly that losses are falling for particles much larger than the skin depth and we expect that the system would indeed become lossless as $d/\delta_s \rightarrow \infty$.

5. (The serious one.) Equation (5) generalizes the Maxwell Garnett formula for elliptical cylinders, introducing an extra parameter Λ . Λ is connected to the shape of the ellipse as square root of the axis ratio, and it is derived in the supplementary materials. In my opinion the correct Λ should be simply the axis ratio, not its square root. Correct me if I am wrong: in terms of the depolarization factor of an ellipsoid N (which for a sphere is $1/3$ and a circle $1/2$), this parameter Λ is $(1-N)/N$, and the depolarization factor can be calculated from the axes lengths of the ellipsoid (see Sihvola: *Electromagnetic mixing formulas and applications*, IET, 1999). For one of the ellipsoid axes being much larger than the other two, N becomes $L_2/(L_1+L_2)$ which leads to Λ being L_1/L_2 , and not its square root.

We are very grateful to the reviewer for noticing this mistake. The parameter Λ should indeed be defined as $\Lambda = L_x/L_y$, not $\Lambda^2 = L_x/L_y$, and we have fixed this in the main text and the supplementary material.

Although Λ was defined incorrectly, we believe that the rest of the derivation is correct and the desired formula is unchanged. However, many readers would

Figure 4: The effective index of distorted square arrays of gold nanocylinders, calculated numerically (solid lines) and with the MG mixing formula using the corrected definition of $\Lambda = L_x/L_y$. The surface-to-surface separation can be distorted (left) or kept fixed (right). If the gaps are distorted then at high aspect ratios the gaps can become very small, which enhances particle-particle interactions and reduces tunability. The agreement with the MG mixing formula is therefore best when the gaps are not distorted.

find the reviewer’s approach of using depolarisation factors more appealing than our lengthy derivation, so we now also include a mention of this approach with a reference to the suggested book.

In figure 4a (now figure 5a) of the main text we had plotted the MG mixing formula for ellipsoids using the incorrect definition of Λ . In figure 4 of this document, we show the corrected formula for two different cases.

In the first case, both the cylinders and the gaps between cylinders are distorted (as was originally the case in the submitted paper). For high aspect ratios the gaps become very small and the particle-particle interactions are very strong, such that the MG mixing formula is not very accurate. In fact, the particle-particle interactions act to reduce the tunability of the system.

In the second case, the gaps between the cylinders are fixed and only the cylinders are distorted (one can imagine, for example, that the spacing is achieved by coating the cylinders with a uniformly thick coating). We see that there is much better agreement with the MG mixing formula and the effective can be tuned much more easily.

We have decided to include only the second case (fixed gaps, distorted cylinders) as a figure of the main paper, but have noted in the main text that for small gap sizes the particle-particle interactions will reduce the tunability and that this can happen when both the gaps and cylinders distorted.

Summary of changes

Line numbers refer to the positions in the revised document.

1. Figure 2 has been split into two figures. For simplicity, all the metals in the new figures 2a-2c are now titanium, and all the metals in the new figures 3a-3e are now gold.
2. The experimental data (previously figure 2h, now figure 3b) has been replotted over a smaller wavelength range to avoid confusion arising from the transmission window of the detector. Consequently, discussion of the detector cut off has been removed from the main text.
3. Higher resolution images of the experimental samples have been added to the supplementary material.
4. The term ‘germanium window’ has been replaced by ‘germanium substrate’.
5. We have added the following new sections to the supplementary material,
 - “Weak magnetic responses”, referenced in the main text at line 112
 - “Origins of the dispersion at near-to-mid infrared wavelengths”, referenced in the main text at line 170
 - “Losses in the telecom and visible wavelengths”, referenced in the main text at line 170
 - “Accuracy of the Maxwell-Garnett mixing formula for square and triangular lattices”, referenced in the main text at line 263
6. We have added descriptions of how the curves in figure 4b (previously figure 3b) were produced at lines 189-193, lines 206-211, and to the caption of the figure.
7. We have amended the reference on line 256 to point to figure 4b.
8. Breaks were added to the x -axis of figure 4b (previously figure 3b), to emphasise the that the solid lines are approaching but do not reach the limiting cases of $d/\delta_s \rightarrow 0$ and ∞ . Lines 201-205 were also changed to add extra emphasis.
9. We have corrected the definition of $\Lambda = \sqrt{L_{\parallel}/L_{\perp}}$ to $\Lambda = L_{\parallel}/L_{\perp}$ in the main text and supplementary material.
10. We have noted that the Maxwell Garnett mixing formula for elliptic cylinders could also be derived using depolarisation factors, and included a reference to *Sihvola, IET, 1999* at lines 226-230.

11. Figure 5a (previously figure 4a) has now been plotted using the corrected definition of Λ . The figure has also been changed to keep the surface-to-surface separation of the cylinders constant. The figure caption and comments on the tunability of the system have been revised accordingly (lines 232, 235, and 249-254).
12. We have raised the maximum amplitudes shown in the near-field plots of figure 6c (previously figure 5c) and included a colorbar scale.

Reviewers' Comments:

Reviewer #1:

Remarks to the Author:

I can see a significant improvement in the quality of manuscript, after authors revised their work. Now it is clear, straightforward, and easy to read. I believe, this work deserves to be published in Nature Communications right after meeting following tiny remarks:

- 1) Page 3, left column, lines 132-138: first it mentions arrays of nanospheres and refers to Figure 2b. Then it says "these arrays ... figure 2c", but data in figure 2c corresponds to arrays of rectangular pillars. Therefore, I believe, it is better to reword "these arrays" to, for example, "such arrays".
- 2) Same page and column, lines 142-143: here it mentions germanium window, while in response to Reviewer 2 authors claimed to change 'window' to 'substrate' in the text.

Reviewer #2:

Remarks to the Author:

The authors have revised their manuscript satisfactorily.

Manuscript NCOMMS-18-33562B

March 27, 2019

Response to Reviewers

We have answered and addressed in full all of the referee's comments.

Reviewer 1

I can see a significant improvement in the quality of manuscript, after authors revised their work. Now it is clear, straightforward, and easy to read. I believe, this work deserves to be published in Nature Communications right after meeting following tiny remarks:

1) Page 3, left column, lines 132-138: first it mentions arrays of nanospheres and refers to Figure 2b. Then it says "these arrays ... figure 2c", but data in figure 2c corresponds to arrays of rectangular pillars. Therefore, I believe, it is better to reword "these arrays" to, for example, "such arrays".

We agree that "such arrays" is more accurate, and have changed the manuscript accordingly.

2) Same page and column, lines 142-143: here it mentions germanium window, while in response to Reviewer 2 authors claimed to change 'window' to 'substrate' in the text.

We agree that this should have been "substrate" not "window" and have now changed the manuscript accordingly.

Reviewer 2

The authors have revised their manuscript satisfactorily.